# The Reliable, Automatic Classification of Neonates in First-Tier MALDI-MS Screening for Sickle Cell Disease

**DOI:** 10.3390/ijns5030031

**Published:** 2019-08-31

**Authors:** Marven El Osta, Pierre Naubourg, Olivier Grunewald, Gilles Renom, Patrick Ducoroy, Jean Marc Périni

**Affiliations:** 1Biomaneo, 22B boulevard Winston Churchill, F-21000 Dijon, France; 2Newborn Screening Laboratory, Biology and Pathology Center, Lille University Medical Centre, F-59000 Lille, France

**Keywords:** sickle cell disease, MALDI-MS, automatic classification, software, β-thalassemia, high-throughput SCD screening, Laboratory Information Management System (LIMS)

## Abstract

Previous research has shown that a MALDI-MS technique can be used to screen for sickle cell disease (SCD), and that a system combining automated sample preparation, MALDI-MS analysis and classification software is a relevant approach for first-line, high-throughput SCD screening. In order to achieve a high-throughput “plug and play” approach while detecting “non-standard” profiles that might prompt the misclassification of a sample, we have incorporated various sets of alerts into the decision support software. These included “biological alert” indicators of a newborn’s clinical status (e. g., detecting samples with no or low HbA), and “technical alerts” indicators for the most common non-standard profiles, i.e., those which might otherwise lead to sample misclassification. We evaluated these alerts by applying them to two datasets (produced by different laboratories). Despite the random generation of abnormal spectra by one-off technical faults or due to the nature and quality of the samples, the use of alerts fully secured the process of automatic sample classification. Firstly, cases of β-thalassemia were detected. Secondly, after a visual check on the tagged profiles and reanalysis of the corresponding biological samples, all the samples were correctly reclassified without prompting further alerts. All of the samples for which the results were not tagged were well classified (i.e., sensitivity and specificity = 1). The alerts were mainly designed for detecting false-negative classifications; all the FAS samples misclassified by the software as FA (a false negative) were marked with an alert. The implementation of alerts in the NeoScreening^®^ Laboratory Information Management System’s decision support software opens up perspectives for the safe, reliable, automated classification of samples, with a visual check solely on abnormal results or samples. It should now be possible to evaluate the combination of the NeoSickle^®^ analytical solution and the NeoScreening^®^ Laboratory Information Management System in a real-life, prospective study of first-line SCD screening.

## 1. Introduction

Newborn disease screening programs are front-line public health measures, and as such must be based on robust analytical methods and data-processing software. Cost effectiveness is a further requirement and has prompted the implementation of high-throughput screening units that reduce unit costs. Lastly, the greatest possible use of automation enables the medical team to focus exclusively on abnormal samples. Our MALDI-MS facility and the associated data processing and interpretation software packages were designed to address these challenges.

Quantitative analysis with MALDI-TOF MS was initially viewed as implausible and inherently irreproducible because the mass spectrum’s signal intensity varies as a function of the sample’s composition, the sample’s morphology, the laser conditions, and sample depletion during continuous laser exposure [1,2]. Inhomogeneous sample deposition is also a crucial obstacle, and leads to poor signal reproducibility (including the so-called “sweet spot” and “coffee-ring” effects) [3,4]. In order to perform a quantitative analysis with the highest possible accuracy, it is important to find the optimal sample preparation method and to perform standard calibrations. Many sample preparation methods have been reported for improving the quality of sample deposition and obtaining fine and uniform crystals: Rapid or slow evaporation [5,6], seeded layers [7], sandwiches [8], and electrospray deposition [9], for example. The chosen method depends on the nature of the target molecules, the instrument used, the MS method applied, and the degree of precision required to answer the analytical question. The use of the MALDI-MS approach in routine clinical practice requires all these parameters to be taken into account.

Hachani et al.’s pilot study was the first to demonstrate the MALDI-MS technique’s potential in sickle cell disease (SCD) screening [10]. A more comprehensive study by Naubourg et al. then demonstrated the relevance of the NeoSickle^®^ approach (i.e., automated, standardized sample preparation, followed by MALDI-MS analysis) and the NeoScreening^®^ sample processing and classification software [11]. Naubourg et al.’s results showed that most of the samples had been correctly classified; however, there was still room for improvement, with a view to obtaining a completely automated, safe tool that enables technicians and biologists to validate results much more rapidly.

Some MS spectra are difficult to interpret directly by the classification software, and must be inspected visually by the operator. After a visual check, the operator can validate (or not) the results generated automatically by the software. There are many reasons why a spectrum may be difficult to interpret: Poor-quality sample collection, processing and/or transport, late sampling, very or extremely preterm infants (as defined by the WHO [12]), low HbA levels, or the presence of atypical Hb variants.

In various domains, software programs have been developed to analyze, process and interpret MALDI MS data [13,14]. However, software solutions for clinical applications of MALDI MS are rare. The companies Bruker Daltonik GmbH (Bremen, Germany) and Biomérieux (Lyon, France) have respectively developed Biotyper^®^ and Vitek-MS^®^ software packages for facilitating and accelerating the identification of microorganisms on the basis of MALDI MS data [15,16]. These programs compare the sample’s spectrum with a database of reference spectra for a range of microorganisms. In the field of infectious disease, PILOT.4lab software (Info Partner SAS, Vandoeuvre-Les-Nancy, France) makes it possible to compile data (including MALDI MS data) from various analytical sources. In order to make a technology like MS usable in the analytical laboratory, it is essential to combine the instruments with data interpretation software and a Laboratory Information Management system (LIMS). The NeoScreening^®^ LIMS from Biomaneo (Dijon, France) meets this objective by including a module (Neoclinical^®^) that interfaces with clinical and/or epidemiological databases, and a module (e-NeoSickle^®^) for the processing and analysis of clinical MALDI-MS data.

It is essential that defective analytical procedures do not lead to misclassifications. An excessively weak signal might prompt an erroneous classification (e.g., HbA low or FAS, rather than FA). Similarly, an undetected blood transfusion (giving an HbA peak that is much larger than the HbS peak) might lead to a classification of FA rather than FAS. In order to achieve a high-throughput “plug and play” approach while detecting “non-standard” profiles that might prompt a misclassification, we have introduced alerts into our decision support software. These alerts tell the operator (a technician or a biologist) that he/she must visually check the classification generated by the software. Our objective is to classify and automatically validate 99% of the FA samples; hence, the operator will only have to check 1% of the samples. A technician can check on technical problems (e.g., an atypical sample from a very premature child, a child with β-thalassemia or a child having been transfused), and a biologist can validate a FAS vs. FS sample. At present, 85% of FA or FAS phenotypes can be validated directly; only 15% are considered to be “non-standard”. The alerts were not designed for FS neonates because all FS profiles should be checked visually.

The objective of the present study was to evaluate the effectiveness of these alerts. We found that the implementation of alerts improved the NeoScreening^®^ LIMS’ reliability. Even though the frequency of non-standard profiles will gradually fall as analytical improvements are made, it is necessary to have tools capable of detecting classification errors related to a one-time specific analytical problem—regardless of how rare the latter may be. In this way, the preconceived idea that MALDI-TOF is a non-reproducible technique might give way to its acceptance as a routine laboratory procedure for application in newborn SCD screening.

## 2. Materials and Methods

### 2.1. Data Collection

The data came from a previously described study of 6701 biological samples [11]. We analyzed residual blood spots from standard Guthrie cards (used in our laboratory’s routine screening activity) for samples with full datasets (i.e., with clinical data, and avalidated Hb phenotypes using reference methods—the French neonatal screening program for SCDs was set up in 1995 under the aegis of the Association Française pour le Dépistage et la Prévention des Handicaps de l’Enfant [17]; the neonatal two-tier SCD screening strategy developed in France combines two of the following technics: Isoelectrofocusing (IEF), capillary electrophoresis (CE), or high performance liquid chromatography (HPLC) [18,19]. In order to select our cohort of Hb phenotyped newborns, we have applied in our laboratory in first line the automated CE system Capillarys^®^ 2 (Sebia, Evry, France) and in second line the automated HPLC system VARIANT nbs (Biorad, Hercules, CA, USA). These technics were applied in the respect of procedures given by the manufacturers. All samples defined as FA, FAC, FAE, FAO-Arab, FAD, FAKorle-Bu, FAX, FC, FE, or F OArab using conventional methods were grouped into a single “FA” class. All samples for which the reference method showed an HbS chain (FAS, FCS, FES, and FS O Arab) were grouped into an “FAS” class. FS homozygotes were defined as “FS”.

There were 71 FS samples, 2919 FAS samples, 3696 FA samples, and 15 S-β+ samples. Respectively 222 and 57 samples in the FA and FAS groups came from very or extremely premature newborns (i.e. born after less than 30 weeks of gestation), and respectively 874 and 249 samples in the FA and FAS groups came from premature newborns (born after 30 to 33 weeks of gestation).

### 2.2. Sample Processing and Analysis

All samples were analyzed at two MALDI-TOF facilities (the University of Burgundy’s CLIPP facility (Dijon, France) and Lille University Hospital’s neonatal screening laboratory (Lille, France)), according to the previously described procedure [11]. At both facilities, samples were prepared for MS measurements using a research version of the NeoSickle^®^ kit (Biomaneo). Samples were deposited in quadruplicate on a 384-spot polished steel MALDI target (Bruker Daltonik GmbH). At both facilities, MS was performed with a MALDI-TOF system: An AutoFlex™ Speed with a 2000 Hz Smartbeam™ II laser (Bruker Daltonik GmbH) in Dijon, and an AutoFlex™ III with a 200 Hz Smartbeam^™^ laser (Bruker Daltonik GmbH) in Lille.

### 2.3. Data Processing

Mass spectrometry acquisitions were analyzed with the algorithm if (i) the whole spectrum and the region of interest were sufficiently intense, (ii) the baseline was not too noisy, and (iii) at least three of the four profiles per sample could be interpreted automatically.

### 2.4. Analytical Data Flow

The NeoScreening^®^ algorithm automatically discriminates between normal samples and samples containing an HbS variant. All the analytical results were centralized via a secure data collector. The newborns’ profiles were automatically classified as FA, FAS or FS. Furthermore, the NeoScreening^®^ software included several alert features.

The first set of alerts (referred to hereafter as “alert set 1”) detected an abnormally low HbA:HbF ratio. The normal range of β:γ and βA:βS ratios was established by reference to the HbA:HbF value given by capillary electrophoresis (CE). The HbA:HbF thresholds were adjusted so that each sample from an infant requiring close monitoring or specific medical care (e.g., very or extremely premature infants) was recorded by the automated classification software.

Alert set 2 was designed to detect border-line MS profiles that cannot be interpreted automatically and reliably by the software. Various indices were used, depending on the parameters to be taken into account.

Alert indexParameter taken into account in the alert“a”Low Beta chain peak intensity“b”Ratio intensity 15,837/15,867 *m*/*z* peak“c”Intensity of Beta chain peak“d”Low HbS chain peak intensity“e”Low HbS chain peak resolution“f”resolving power of the HbS and HbA peaks“g”Ratio HbS/HbA peaks

There were no alerts for newborns with S-thalassaemia (Sβ^+^ or Sβ^°^), since these samples were always classified as FS by the NeoSickle^®^ software [11].

### 2.5. The Data Collector

Sickle Cell Anemia Collect and Compare (SCACC) is a web application (Biomaneo) that helps biologists to compare MS screening results with those of reference screening methods (CE and HPLC). The application stores, groups and presents heterogeneous data in a user-friendly way, including clinical data on the sample donor, the experimental data from CE and HPLC analyses, the experimental data from the MS analysis, and the validation files (i.e., the screening results sent to the pediatrician). SCACC contains a table in which all the information available for a given sample is shown on a single line. Along with a filter system for each variable, this layout makes it easy to create pools of interest (preterm samples, pathologic results, etc.) for the analysis of any misinterpreted results.

### 2.6. Visual Assessment of MS Profiles

Spectra were classified as “standard” and “non-standard”, depending on their quality [11]. The MS profiles were considered to be “non-standard” for one of three reasons. Firstly, some profiles had a non-regular baseline or a slightly distorted peak; this applied to (i) FA MS profiles with a variably broad/sharp/distorted peak but very low intensity at 15,837 *m*/*z* ± 5 Da, relative to the βA peak; and FAS MS profiles with a slight deformation of the βS peak. Secondly, some profiles had a low, broad peak at 15,837 *m*/*z*; this applied to (i) FA MS profiles with a variably shifted peak, and (ii) FAS MS profiles characterized by a broad, well-centered peak at 15,837 ± 5 *m*/*z*. Thirdly, some FA MS profiles had a broader β peak that overlapped to a variable extent with the region of interest at 15,837 *m*/*z* (and thus led to misclassification as “FAS”), whereas other FAS MS profiles showed low resolution and thus poor separation of the β and βS peaks—giving a single, large peak as a shoulder, and no plateau between the peaks.

## 3. Results

### 3.1. Detection of FA Samples Containing Low Amounts of HbA

Alert set 1 detected 2 and 6 samples with an abnormally low amount of HbA (relative to the total amount of Hb) at the Dijon and Lille analytical facilities, respectively. The alert set’s threshold was determined so that samples with a low amount of HbA were always flagged up. A subsequent HPLC analysis of the 2 samples tagged in Dijon revealed β°-thalassemia in both cases. A subsequent HPLC analysis of the 6 samples tagged in Lille revealed (in addition to the 2 cases of β°-thalassemia) 4 samples defined as “low HbA” by the thresholds established in our laboratory (Table 1).

The threshold for the alert set 1 enabled the biologist to provide additional information (i.e., as well as the FA/FAS/FS classification). These spectra were also tagged by alert set 2 (see below).

### 3.2. Frequency of the Profiles Tagged by Alert Set 2 (FA Samples)

Alert set 2 was intended to detect borderline MS profiles that the software could not interpret reliably.

Table 2 shows how the FA samples were interpreted by the reference methods. It is important to note that all the “standard” MALDI spectra were correctly classified, regardless of the analytical center. The software misclassified 22 “non-standard” spectra in Dijon and 7 in Lille. All the samples were tagged, enabling the operators to reclassify the spectra after a visual check (Table 2). A total of 113 correctly classified spectra were confirmed visually: 27 “standard” spectra (7 in Dijon and 20 in Lille) and 86 “non-standard” spectra” (56 in Dijon and 30 in Lille).

Overall, 2.3% of the spectra in Dijon and 1.5% of the spectra in Lille were tagged by an alert. These results are in line with the quality of the data produced at each facility [3]. Although less than 2.5% of the spectra required visual confirmation, this process enabled the correct classification of all the spectra automatically misclassified by the software in Dijon or in Lille.

### 3.3. Frequency of Alert Types for FA Samples

Figure 1 shows the frequencies of the various alerts at each analytical facility and as a function of the quality of the data (“standard” vs. “non-standard” spectra).

Most of the “standard” spectra classified as FA by the MALDI-MS approach were marked by the “FA-a” alert (Figure 1A), corresponding to a weak β chain signal (Figure 2B).

The FA-b alert (Figure 1B) tagged 66% of the “non-standard” spectra classified as FA in Lille and 80% of those in Dijon. This alert is based on the signal intensity ratio of the peaks at 15,837 *m*/*z* and 15,867 *m*/*z*, and thus corresponds to spectra with an abnormal β:βS intensity ratio (Figure 2C). The FA-c alert (based on the resolution of a putative βS peak (Figure 2D)) tagged just one of the 3161 “standard” FA spectra in Dijon and none of the 3386 “standard” FA spectra in Lille.

The FAS-a and FAS-g alerts tagged more than 90% of the misclassified spectra (i.e., samples classified as FA by the reference method but classified as FAS by MALDI MS); these alerts corresponded to spectra with either a weak βS peak (Figure 1C) or an abnormal β:βS intensity ratio.

### 3.4. Frequency of Profiles Tagged by Alert Set 2 (FAS Samples)

Table 3 shows how FAS samples were interpreted by the reference methods. It is noteworthy that all the “standard” MALDI spectra were correctly classified at the Lille facility. The 16 “standard” samples misclassified at the Dijon facility were tagged by an alert for visual checking. The operator was then able to reclassify the sample or reanalyze it using the same method or a complementary approach (HPLC, CE, etc.), and thus avoided any final classification errors.

The software misclassified 22 “non-standard” spectra in Dijon and 30 in Lille. However, all these spectra triggered an alert, which allowed the operators to correct the automated classification (Table 3).

Considering Dijon and Lille together, a total of 670 correctly classified “standard” spectra were marked with an alert; a visual check confirmed that they had been correctly classified. One hundred and twenty-seven “non-standard” spectra with a correct FAS classification were visually confirmed. 22% of the spectra in Dijon and 8.3% of the spectra in Lille were tagged with an alert. Again, these results are in line with the quality of the data produced at each facility [3]. Therefore, fewer than 25% of the spectra classified as FAS by the reference methods were tagged for visual confirmation, which enabled the operator to correct all of the spectra misclassified automatically by the software. The alert rate of ~25% for FAS samples is not unusual, given that these samples must be closely inspected prior to validation with a second analytical method.

### 3.5. Frequency of Alert Types for FAS Samples

Figure 3 shows the frequencies of the various alerts at each analytical facility and as a function of the quality of the data (“standard” vs. “non-standard” spectra).

The majority of “standard” or “non-standard” spectra correctly classified as FAS triggered a FAS-g alert (Figure 3A,B), corresponding to an out-of-range βS:βA intensity ratio (Figure 4E). In such a case, the operator was able to immediately check the spectrum for the presence or absence of a βS chain peak. Although the FAS-g alert was predominant, it nonetheless represented less than 8% of all the spectra generated from FAS samples—most of which had a “standard” spectrum (Figure 4A).

The FAS-f alert was the second most frequent alert for correctly classified “standard” FAS spectra in Dijon (Figure 3A); this corresponded to inability to resolve the βS and βA peaks (Figure 3D). Very few spectra classified as FAS were tagged by the FAS-d and FAS-e alerts, based respectively on the intensity (Figure 4B) and the resolution (Figure 4C) of the βS chain peak.

In Dijon, 16 “standard” spectra samples classified as FAS by the reference methods were incorrectly classified as FA by the MALDI MS method. Figure 3C shows that these 16 spectra were associated with an FA-b type alert, due to an abnormal β:βS intensity ratio (Figure 2C). Figure 3D shows that the 22 “non-standard” FAS samples incorrectly classified as FA in Dijon and the 30 such samples in Lille were tagged with an FA-b alert; again, this enabled the operators to correct the classification indicated by the software.

An analysis of all the FA and FAS samples showed that the two main alerts were FA-b (*n* = 152) and FAS-g (*n* = 405). Both of these alerts are based on the βS:βA ratio.

We did not observe a correlation between the type of alert and the various criteria (developed previously [3]) used to define a “non-standard” spectrum.

### 3.6. Comparison of the Analytical Facilities with Regard to the Frequency of the Alerts

The frequencies of the FA alerts were essentially the same in Dijon and in Lille. All the alerts for misclassified FAS samples were FA-b, and most of the alerts for correctly classified FA samples were FA-a. The intercenter differences were greater for alerts associated with FAS samples; for example, the FAS-f alert (corresponding to a lack of resolution) was more frequent at the Dijon analytical facility (for 174 standard FAS spectra) than at the Lille facility (4 spectra). This difference is consistent with the lower overall quality of the results in Dijon.

We have already shown that when a profile was misclassified at one MS facility, the corresponding profile at the other MS facility was correctly classified and was less likely to be a non-standard profile [3]. These occasional misclassifications were mainly due to a poor-quality spectrum, which in turn was due to technical incidents such as (i) tip blockage and thus non-deposition of the sample or (ii) an off-axis tip and thus a sample that was poorly centered on the MALDI target. Here, the effect on the alert was the same; although FA-a, FA-b, and FAS-f alerts arose in both facilities, they were not associated with the same spectra. Indeed, only a few samples were tagged by the same alert at both facilities; these cases were related to the biological nature of the sample itself, rather than a technical problem.

### 3.7. Alerts Triggered by Samples from Newborns with Thalassaemia

The e-NeoSickle^®^ classification algorithm was developed to classify biological samples into 4 categories: FA, FAS, FS, and uninterpretable. Major beta-thalassemic samples that do not meet any of these criteria are inevitably misclassified but must always be flagged up. Our results show that one of the beta thalassemic samples was classified as FS; this type of sample was always checked using a second method. The second beta thalassemic sample was classified as “FAS” and was tagged by an FAS-a alert at both facilities. The ensuing visual check enabled the biologist to reclassify the sample as β-thalassemia major.

Thalassemia beta samples were detected using several approaches: The FS classification, the FAS-d/a alert, and alert set 1. A forthcoming software upgrade should lead to the direct classification of these samples as beta-thalassemia major.

### 3.8. Comparison of Alert Frequencies and Misclassification Frequencies

All the non-tagged profiles were well classified. In the group of tagged profiles classified as FA by the algorithm, only 24.7% and 12.3% of the samples (in Dijon and Lille, respectively) were misclassified (they were FAS phenotypes, according to the conventional methods). In the group of tagged profiles classified as FAS by the algorithm, only 6.1% and 12.5% of samples (in Dijon and Lille, respectively) were misclassified (they were FA phenotypes, according to the conventional methods). No misclassified profiles were not tagged by alerts. 

These results show that the alert thresholds are very strict. An increase in the number of analyses and the continuous improvement of the procedures should make it possible to decrease the frequency of both alerts and misclassified samples.

## 4. Discussion

Neonatal screening for SCD must be trustworthy under all circumstances. To avoid phenotyping errors, insufficiently reliable results must be identified and tagged with an alert. This requirement is justified by the nature of neonatal screening, which is at the forefront of preventive public health measures.

The NeoScreening^®^ software was developed to make the interpretation of SCD screening results as easy and fast as possible. To this end, the results are color-coded so that operators can instantly visualize the samples’ classifications: The samples classified as FA appear in green, those classified as FAS samples appear in orange, and those classified as FS appear in red. For samples that have triggered an alert, a question mark is used to attract the operator’s attention.

Several criteria have been programmed for the detection of a non-standard spectrum that may (depending on their phenotype) cause the corresponding sample to be misclassified. Hence, several types of non-standard spectra have been defined. The worst error for automatic classification is the assignment of an FA phenotype to an FAS newborn. The automatic discrimination of “standard” and “non-standard” profiles (on the basis of CE results) was monitored for alerts; abnormal curves are displayed in red [17].

As mentioned above, there are two types of alert in the NeoScreening^®^ software: “Technical” alerts are related to low-quality data, whereas “biological” alerts are related to the characteristics of the sample itself (late sampling, extreme prematurity (as defined by the WHO), low HbA, atypical variants, etc.). The variables taken into account differ as a function of the type of alert. Low β/γ and β/βS ratios were used as markers of a neonate’s clinical status, rather than as quality controls.

There are two main reasons for adding alerts to the automatic classification of SCD phenotypes. The first is low signal intensity at 15,837 and/or 15,867 *m*/*z*. In this case, the signal measurement takes background noise into account, despite prior processing of all the MS data (i.e., the subtraction of random (electrical) noise and baseline (chemical) noise). A noisy background leads to an irregular baseline that can even mimic low, wide peaks in some high-resolution profiles. Secondly, poor resolution of the β and/or βS chains can result not only from poor adjustment of the instrumentation but also from the analysis of four combined (i.e., averaged) spectra. Our previous research [11] highlighted the limitations of quadruplicate analysis as part of a MALDI-MS analytical solution.

Although the alert triggering thresholds were designed for use at two analytical facilities equipped with the same type of MS system (an Autoflex IV in Dijon and an Autoflex III in Lille), the facilities differed with regard to the frequencies of the various alerts and the types of spectra that triggered the alerts. However, these dissimilarities emphasize that it was possible to obtain an interpretable result for all the samples; the occasional absence of a MALDI MS result therefore can be ascribed to one-off defects and “non-standard” spectra that are nevertheless tagged by “technical” alerts.

We would expect optimization of the sample processing and mass measurement steps to further reduce the number of samples (other than those from extremely premature infants) with abnormally low signal intensities.

We chose not to program alerts for FS neonates because all FS profiles (homozygous SS samples or Sβ+ or Sβ° samples) must be checked visually and then confirmed using a reference method.

However, samples for which the spectra were classified FAS in the absence of an alert were also re-analyzed; this is an obligation under France’s legislation on SCD screening [18].

Samples for which the spectra are classified as FA in the absence of an alert do not undergo further analysis; hence, none of the FS or FAS samples must be included in this category. In the present study, all the samples classified as FA in the absence of an alert by the MALDI MS system were also classified as FA by the reference methods.

In the future, we shall study the FAS and FA spectra that triggered alerts from set 1 in more detail because the latter concern the characteristics of the biological sample per se (extreme prematurity, transfused infants, infants with beta thalassemia, low HbA levels, etc.). After analysis with the NeoSickle^®^ solution, these samples’ biological characteristics will be systematically characterized and confirmed using an additional analytical method.

All the FAS samples misclassified by the software as FA were tagged with an alert; an initial visual verification and then sample retesting enabled correct reclassification in all of these cases.

In the present study, 708 spectra were tagged in Dijon and 298 were tagged in Lille. Considering that all the FAS and FS samples must be reanalyzed, only correctly classified but tagged FA samples (63 in Dijon and 50 in Lille) prompted an unnecessary visual reanalysis. This corresponds to only 0.9% and 0.7% of the samples, respectively.

Designing a set of alerts is a compromise between not validating misclassified spectra on one hand and avoiding an excessively high reanalysis rate on the other. In routine use, the NeoSickle^®^ solution tags suspect spectra and alerts the operator. After a visual check, the operator will validate or invalidate the result initially produced by the software. In the latter case, the operator may decide to reanalyze the same target (to confirm his/her decision) or to reinitiate the entire analytical procedure (i.e., sample processing, deposition and analysis). Given that the present study included a high proportion of FAS, FS, premature or extremely premature newborns, it is difficult to calculate the percentage of samples that will require reanalysis in routine use. Nevertheless, one can extrapolate the data and predict that less than 1% of the FA spectra will be mislabeled. Of these mislabeled spectra, around 3 in 10 will be reclassified solely on the basis of the visual check, 5 in 10 will be reanalyzed on the same MALDI target, and only 2 in 10 (i.e., less than 0.2% of all the samples) will require reinitiation of the entire NeoSickle^®^ analytical procedure or analysis with another approach.

The potential correlation between the different alerts and the variants other than HbS will be studied after we have acquired a large number of spectra from the corresponding samples. It is considered that certain variants of the hemoglobin chains are expressed to different extents, which might enable a quantitative approach to developing alerts for certain variants.

With the inclusion of additional alerts in the NeoScreening^®^ LIMS’s decision support software, all misclassifications reported in our previous article [11] were detected and tagged by an alert. We showed that the mathematical data processing model and the classification algorithm (which take account of variables such as the signal-to-noise ratio, the intensity and the resolution) make it possible to produce reliable results even when “degraded” spectra (according to visual criteria) are present. We now intend to test the automatic MALDI-TOF classification strategy for first-line SCD screening in a prospective, real-life study.

## Figures and Tables

**Figure 1 IJNS-05-00031-f001:**
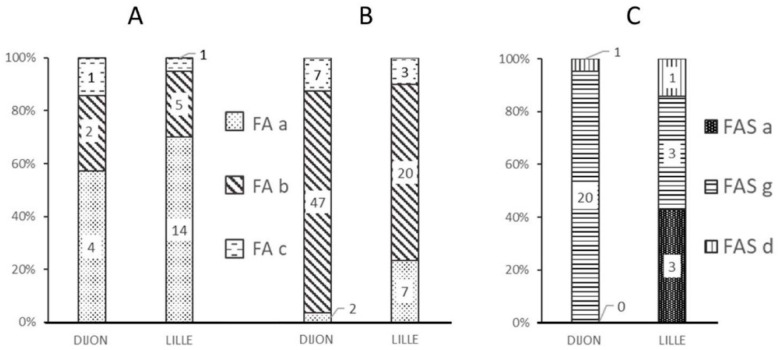
Frequency of the different alerts (FA-a, FA-b, FA-c, FAS-a, FAS-g, and FAS-d) at each analytical facility and as a function of each spectrum’s MALDI-MS classification. (**A**) Alerts for “standard” MALDI FA spectra; (**B**) Alerts for “non-standard” MALDI FA spectra; (**C**) Alerts for misclassified “non-standard” MALDI spectra.

**Figure 2 IJNS-05-00031-f002:**
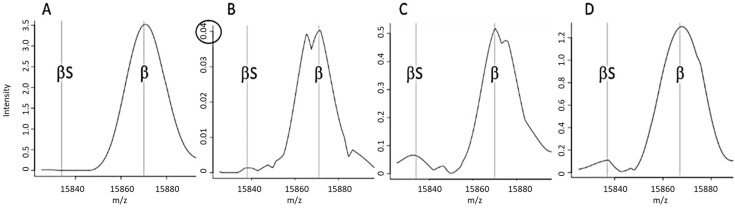
Examples of spectra tagged by a particular alert. (**A**) A “standard” FA spectrum that did not trigger an alert, (**B**) A spectrum tagged by an FA-a alert (an abnormally weak β chain peak), (**C**) a spectrum tagged by an FA-b alert (an abnormal β:βS intensity ratio), (**D**) A spectrum tagged by an FA-c alert (resolution of a putative peak for abnormal βS).

**Figure 3 IJNS-05-00031-f003:**
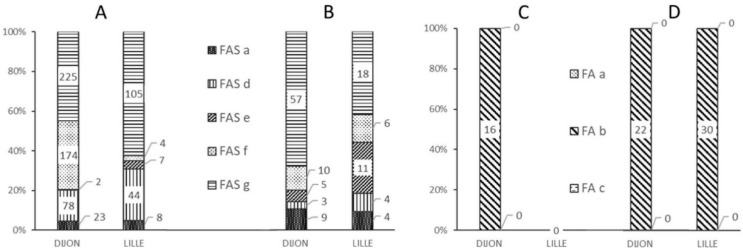
Frequency of different alerts as a function of the analysis center and the MALDI-MS classification of the spectrum. (**A**) Alerts for “standard” FAS MALDI spectra, (**B**) Alerts for “non-standard” FAS MALDI spectra, (**C**) Alerts for misclassified “standard” FA MALDI spectra, (**D**) Alerts for misclassified “non-standard” FA MALDI spectra.

**Figure 4 IJNS-05-00031-f004:**
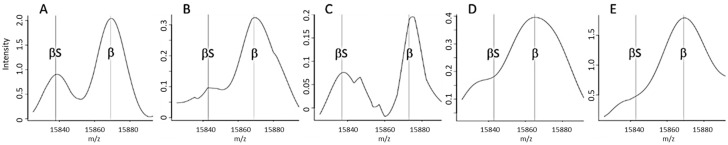
Examples of spectra tagged by an alert from set 2. (**A**) Standard FAS spectrum without an alert, (**B**) A spectrum tagged by an FAS-d alert: abnormally weak signal intensity for the βS chain, (**C**) A spectrum tagged by an FAS-e alert: insufficient resolution of the βS peak, (**D**) A spectrum tagged by an FAS-f alert, indicating an insufficient resolution of the βS and βA peaks, (**E**) Spectrum tagged by an FAS-g alert, corresponding to an abnormal βS:βA peak ratio.

**Table 1 IJNS-05-00031-t001:** Frequency of the alert set 1, as a function of the birth term. w.a.: Weeks of amenorrhea.

	Lille			Dijon		
Birth term (w.a.)	<33	33–36	>36	<33	33–36	>36
Frequency	1/222 (0.5%)	2/874 (0.2%)	3/2600 (0.1%)	0/222 (0%)	0/874 (0%)	2/2600 (0.1%)

**Table 2 IJNS-05-00031-t002:** Frequency of alert set 2 for FA samples at each analytical facility.

**Classification and Alerts Generated by the Algorithm**	**Visual Quality of the Spectra**		**Corrected FA (Classified by Reference Methods)**			
			**Dijon (3627)**		**Lille (3675)**	
			**S**	**NS**	**S**	**NS**
	FA	no alert	3154	389	3366	252
		with an alert	7	56	20	30
	FAS	no alert	0	0	0	0
		with an alert	0	21	0	7

**Table 3 IJNS-05-00031-t003:** Frequency of alert set 2 for FAS samples, according to the analysis facility.

**Classification and Alerts Generated by the Algorithm**	**Visual Quality of the Spectrum**		**Corrected FAS (Classified by Reference Methods)**			
			**Dijon (2853)**		**Lille (2902)**	
			**S**	**NS**	**S**	**NS**
	FA	no alert	0	0	0	0
		with an alert	16	22	0	30
	FAS	no alert	1989	240	2585	76
		with an alert	502	84	168	43

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
