# Peer review of "The Reliable, Automatic Classification of Neonates in First-Tier MALDI-MS Screening for Sickle Cell Disease"

_2409-515X, 2019, doi:10.3390/ijns5030031_

Round 1

Reviewer 1 Report

This manuscript by El Osta et al describes an automated system for the classification of sickle cell disease screening results obtained by MALDI MS analysis. In routine neonatal screening laboratories, sickle cell disease is done by HPLC, CE or IEF. In their previous work (Reference 3), this group described the use of MALDI TOF MS for screening of this disorder. The current manuscript appears to this reviewer as a further improvement and is a step forward. However, the manuscript is not clearly presented and needs significant improvement to clarify what has been done and what was achieved.

Concerns:

1-    Objectives: the authors stated two sets of objectives in different paragraphs (line 77 and line 83). This is confusing and the objectives of the study should be consolidated.

2-    On line 83-84, the authors stated objective of the present study was to evaluate the effectiveness of these alerts in terms of the analysis rate”. The use of “analysis rate” in this context is confusing and should be re-worded.

3-    Authors unusually grouped normal and abnormal hemoglobin patterns (i.e. FA, FC and FE) under one category that they named FA (line 94). That FA is the normal newborn pattern; this experimental design needs to be justified. This also applies to the group designated by the authors as FAS which indeed included abnormal sickling variants such as FCS (line 96). Please justify the logic behind this grouping.

4-    Line 100-101: The authors referred to the gestation ages as “weeks of amenorrhea”. Please convert to standard terminology.  

5-    Under analytical data flow, the paragraph (line 125-134) should be converted from text format to Table format.

6-    What do m/z 15850, 15880 and 15837 represent?

7-    Line 141, SCASS should be SCACC

8-     Online 149, authors use vague terminology that a peak has very low intensity. Standard scientific terminology such as signal to noise ratio should be used

9-    The labels of the x and y-axis of Figure 2 and 4 are not legible. This needs to be corrected.

10- The predictive value of alerts (line 286-301):

The definition of the positive predictive values by the authors in line 288 (i.e. the probability that a tagged sample is an FA sample misclassified as FAS) is not consistent with the known definition in neonatal screening context (i.e. PPV is the likelihood that a presumptive positive is a true positive). Further, the PPV of 24.7% and 12.3% for the two testing sites are highly variable and are both quite low. Please comment

Reviewer 2 Report

The manuscript “The reliable, automatic classification of neonates in first-tier MALDI-MS screening for sickle cell disease” describes the development of the automatic platform based on MALDI-MS towards the screening of the sickle cell disease.

Although, the topic of this research is certainly very interesting, the manuscript is suffering from the lack of scientific novelty and analytical merit it terms of a score of applications. Moreover, the application of MALDI-MS for the quantitative analysis and sometimes even for high-throughput screening still remains a great challenge due to presence of the "sweet spot" phenomenon. A lot of efforts were dedicated to make MALDI‐MS a quantitative technique comparable with the efficiency of liquid chromatography mass spectrometry (LC/MS). However, the preconception of MALDI being a non‐reproducible technique due to the "sweet spot" phenomenon, resulted in MALDI‐MS being never really accepted as routine laboratory procedure. In this regard, the link to the technical and analytical merit of the proposed “decision support software” is unclear. In addition, the manuscript is written in a very vague manner. Below the authors will find a list of suggestions what can help to improve the manuscript on focus and clarity.

1.      The summary (p.1, line 8-31) is too long and is written in non-typical manner. The presented summary suits better to the introduction.

2.       “This reliable, completely safe tool saves the technician and the biologist considerable time when validating the results.” – how the validation of this “reliable tool” was done? Which alternative tools were utilized to validate the obtained results and to verify the proposed platform?

3.      The analytical merit of the reported platform is doubtful so far if “Alerts were not designed for FS neonates because all FS profiles should be checked visually” (see p.1. line 23). The clear methodological algorithm must be proposed then in the manner: “if so, when what…when not so, then what…?” It would make sense to summarize such an algorithm in a graphical scheme, indeed.

4.      p.2, line 54-55: As a main reason resulting in the difficulties to mass spectra interpretation, viz. low signal to noise ratio, presence of the buffer and interfering species, adducts formation, an intensive targeted fragmentation and low qualification of the operators should be highlighted.

5.      p.3, Data processing, „… were analyzed by the algorithm if (i) the whole spectrum and the region of interest were sufficiently intense, (ii) the baseline was not too noisy, and (iii) at least three of the four profiles per sample could be interpreted automatically.” Instead of these vague described criteria, the concrete evaluation criteria must be given, viz. S/N ratio, isotopes ratio, fragmentation degree, etc.

6.       Figure 1, Figure 3, please specify ±SD,  RSD (%).

7.       Figure 2, Figure 4 – given mass spectra are not readable. In addition, it remains unclear this mass was used for (average mass, monoisotopic?) for classification and decision support software.  

8.      The lack of references within this manuscript is obvious.

Round 2

Reviewer 1 Report

The Authors improved their manuscript and addressed all the points that I raised in my initial review.

Author Response

Reviewer 1 did not request any further information.

Reviewer 2 Report

The authors have partly improved their manuscript on focus and clarity, however, the manuscript is still not clearly presented and not all queries are addressed in a fully and convenient way.

The major concern within this submission still remains the unclear described "visual evaluation" of MS profiles. The corresponding MS part remains very weak.

Query 2. No data obtained by means of "reference methods", viz. HPLC-MS or electrophoresis were shown in the manuscript or ESI. Please add these data (if you have). Moreover, in the "Materials and Methods" part, there is no description of the used CE and HPLC assays. By the way, which detector was utilized? If the authors used LC-ESI-MS method, the appropriate separation conditions, type of the column, type and speed of the mobile phase, as well as MS parameters must be specified. The same is for the used CE assay. The data obtained by means of the references methods must be shown in comparison to the novel MALDI-TOF approach.

Query 3. The given clarification should be added into the manuscript.

Query 4. It´s still unclear in the text why the novel technology allowing running the samples in a high-throughput manner can help to avoid the typical difficulties occurring within the mass spectra interpretation.

Query 5. The query 5 was not addressed in the manuscript. Thus, the clarification "we showed that the mathematical data processing model and the classification algorithm (which take account of variables such as the signal-to-noise ratio, the intensity and the resolution) make it possible to produce reliable results even when "degraded" spectra (according to visual criteria) are present." needs to be added into the manuscript to make it clear for the reader.

Query 6. This clarification needs to be added into the manuscript.
